# ACVR1: A Novel Therapeutic Target to Treat Anemia in Myelofibrosis

**DOI:** 10.3390/cancers16010154

**Published:** 2023-12-28

**Authors:** Andrea Duminuco, Helen T. Chifotides, Sebastiano Giallongo, Cesarina Giallongo, Daniele Tibullo, Giuseppe A. Palumbo

**Affiliations:** 1Hematology Unit with BMT, A.O.U. Policlinico “G.Rodolico-San Marco”, 95123 Catania, Italy; giuseppe.palumbo@unict.it; 2Department of Leukemia, The University of Texas MD Anderson Cancer Center, 1400 Holcombe Blvd., Houston, TX 77030, USA; HTChifotides@mdanderson.org; 3Department of Medical, Surgical Sciences and Advanced Technologies “G.F. Ingrassia”, University of Catania, 95123 Catania, Italy; sebastiano.giallongo@unict.it (S.G.); cesarina.giallongo@unict.it (C.G.); 4Department of Biomedical and Biotechnological Sciences, University of Catania, 95123 Catania, Italy; d.tibullo@unict.it

**Keywords:** ACVR1, ACVR1 inhibitor, anemia, hepcidin, JAK inhibitor, myelofibrosis, momelotinib, myeloproliferative neoplasms, pacritinib, ruxolitinib

## Abstract

**Simple Summary:**

The human activin receptor type I (ACVR1) is a complex protein that regulates production of hepcidin on hepatocytes and red blood cells. Hepcidin is a small peptide that regulates iron metabolism and plasma iron levels. High hepcidin levels are associated with anemia, which is a hallmark of myelofibrosis. Myelofibrosis is a cancer of the bone marrow, characterized by fibrosis and anemia, splenomegaly, and systemic symptoms. Anemia and red blood cell transfusions negatively impact prognosis in myelofibrosis. Ruxolitinib and fedratinib (JAK inhibitors) may exacerbate anemia in myelofibrosis patients. Momelotinib and pacritinib are potent ACVR1 inhibitors that are preferable to treat cytopenic patients with myelofibrosis. In September 2023, momelotinib was approved as a treatment for anemic patients with myelofibrosis based on the phase 3 clinical trials SIMPLIFY-1 and MOMENTUM, which demonstrated the marked anemia benefits of momelotinib. Other ACVR1 inhibitors (e.g., zilurgisertib) are evaluated in early phase clinical trials as treatments for anemia in MF patients.

**Abstract:**

Activin receptor type I (ACVR1) is a transmembrane kinase receptor belonging to bone morphogenic protein receptors (BMPs). ACVR1 plays an important role in hematopoiesis and anemia via the BMP6/ACVR1/SMAD pathway, which regulates expression of hepcidin, the master regulator of iron homeostasis. Elevated hepcidin levels are inversely associated with plasma iron levels, and chronic hepcidin expression leads to iron-restricted anemia. Anemia is one of the hallmarks of myelofibrosis (MF), a bone marrow (BM) malignancy characterized by BM scarring resulting in impaired hematopoiesis, splenomegaly, and systemic symptoms. Anemia and red blood cell transfusions negatively impact MF prognosis. Among the approved JAK inhibitors (ruxolitinib, fedratinib, momelotinib, and pacritinib) for MF, momelotinib and pacritinib are preferably used in cytopenic patients; both agents are potent ACVR1 inhibitors that suppress hepcidin expression via the BMP6/ACVR1/SMAD pathway and restore iron homeostasis/erythropoiesis. In September 2023, momelotinib was approved as a treatment for patients with MF and anemia. Zilurgisertib (ACVR1 inhibitor) and DISC-0974 (anti-hemojuvelin monoclonal antibody) are evaluated in early phase clinical trials in patients with MF and anemia. Luspatercept (ACVR2B ligand trap) is assessed in transfusion-dependent MF patients in a registrational phase 3 trial. Approved ACVR1 inhibitors and novel agents in development are poised to improve the outcomes of anemic MF patients.

## 1. ACVR1 Structure and Biological Significance

The human activin receptor type I (ACVR1), also known as activin receptor-like kinase-2 (ALK2), is a member of the bone morphogenic protein (BMP) receptors that belong to the receptors of the transforming growth factor-β (TGF-β) superfamily, which includes activins, BMPs, and growth differentiation factors. The TGF-β superfamily is ubiquitously expressed in human tissues, and TGF-β signaling controls a wide range of biological processes, including hematopoietic stem cells [1]. Several studies have demonstrated the biological significance of ACVR1 in the early stages of development and repair of the skeletal system throughout one’s lifetime [2,3]. BMPs induce osteogenic signaling and play a critical role in developing and repairing bone and cartilage.

ACVR1 is a transmembrane kinase receptor that consists of an extracellular N-terminal ligand-binding domain, a transmembrane (TM) domain, an intracellular glycine–serine-rich (GS) domain, and a protein kinase domain [4,5]. ACVR1 and the type II receptors BMPR2, ACVR2A, and ACVR2B form hetero-tetrameric complexes comprising two type I and two type II receptors. The type II receptors transphosphorylate the GS domain of the type I receptors upon ligand binding, and activate the kinase domain, which phosphorylates the small-mothers-against decapentaplegic (SMAD) 1/5/8 proteins (Figure 1) [1,6].

SMAD intracellular signaling proteins comprise two highly conserved domains: the Mad homolog domain 1 (MH1), which binds DNA, and the Mad homolog domain 2 (MH2), which is crucial in protein–protein interactions; MH1 and MH2 are located at the N-terminus and C-terminus, respectively [7]. Once phosphorylated, SMAD proteins form a complex with the common mediator SMAD4 and migrate to the nucleus, where they bind to different SMAD-binding elements, DNA transcription factors, transcriptional co-activators, or co-inhibitors, ultimately modulating the expression of a plethora of target genes (Figure 1) [8]. SMAD activity is related to chromatin modifiers because SMAD-containing complexes recruit the ATPase subunit of SWI-SNF chromatin remodelers, along with BRG1 and the histone acetyltransferases p300 and CREB binding protein [9,10]. MH1 and MH2 are separated by a proline-rich region, which is phosphorylated by mitogen-activated protein kinases, glycogen synthase kinase-3b, and cyclin-dependent kinases [11]. In the TGF-β family signalling, SMAD-dependent and non-SMAD signaling pathways have been characterized (Figure 1) [12].

## 2. ACVR1 in Human Diseases

The role of ACVR1 in human diseases was first described in fibrodysplasia ossificans progressiva (FOP). FOP, often referred to as “stone man syndrome”, is a remarkably rare (one patient in 2 million) and devastating genetic disorder that profoundly impacts one’s life due to the progressive change of soft tissues, including muscles, ligaments, and tendons, into bone over time, where any trauma or injury of the affected areas triggers a cascade of events leading to heterotopic ossification (HO) [13,14]. FOP is caused by a mutation in the *ACVR1* gene and the subsequently altered BMP functions. The gain-of-function mutation c.617G>A (Arg206His) in the *ACVR1* gene was the first to be reported and is the most common mutation in FOP patients (more than 95% of the cases). In FOP, the mutated receptor ACVR1 is overactive, leading to an abnormal response of soft tissues to injuries or inflammation through dysregulated BMP signaling. Recent findings demonstrated the central role of activin A in activating ACVR1 and, thus, the development and progression of HO. Barruet and colleagues demonstrated that the mutation *ACVR1* R206H increased human induced pluripotent stem cell-derived endothelial cells in FOP patients [15].

Furthermore, the occurrence of neurological symptoms in FOP suggests that ACVR1 is involved in the development and regulation of the central nervous system. Diffuse intrinsic pontine glioma, a lethal childhood brainstem tumor, is a cerebral neoplastic disease in which 25% of the patients harbor somatic mutations in *ACVR1* [16]. BMP signaling is also involved in heart and valve morphogenesis; deletion of *ACVR1* leads to altered expression of genes involved in cardiac differentiation, defective valve formation promoting cardiovascular diseases, and congenital heart defects [17,18].

ACVR1 induces and regulates expression of Sonic Hedgehog (Shh) [19,20] besides affecting hepcidin expression and hemoglobin (Hb) levels in myelofibrosis (MF). Shh is a member of the hedgehog family that induces proliferation of primitive human hematopoietic cells via BMP regulation; our group reported that the Shh axis drives bone marrow fibrosis in primary MF [21]. Also, Bhardwaj and colleagues showed that Shh stimulation enhanced primitive pluripotent human hematopoietic cell expansion and that a specific inhibitor of BMP4 inhibited Shh-induced proliferation as hedgehog antibodies did [22].

Importantly, ACVR1 is associated with hepcidin expression, hematopoiesis and anemia via the BMP/ACVR1/SMAD pathway in MF. This role of ACVR1 is also inferred by the activity of rusfertide, a hepcidin-mimetic that sequesters iron in the reticuloendothelial system, which restricts the availability of iron for erythropoiesis and thus, decreases Hb levels, in patients who have polycythemia vera [23]. Pathogenic mutations in the *TMPRSS6* gene encoding matriptase-2, a liver-specific transmembrane serine protease that plays a central role in downregulating hepcidin, result in uninhibited hepcidin production in the iron-refractory iron deficiency anemia syndrome [24].

## 3. ACVR1 Inhibition

Inhibition of ACVR1/ALK2 is currently studied as a target of several agents in clinical development. Several ACVR1/ALK2 inhibitors have demonstrated efficacy in vivo and preclinical models and are investigated in clinical trials [25].

Importantly, in the past few years, several treatments for hepcidin dysregulation and anemia, including ACVR1/ALK2 inhibitors, were developed [26,27]. ACVR1/ALK2 inhibitors have been of great interest in the treatment of MF (discussed below) after discovering that momelotinib [28] and, more recently, pacritinib [29] inhibit ACVR1/ALK2 and the hepcidin–ferroportin axis, resulting in notable anemia benefits.

## 4. Hepcidin

In mammals, iron homeostasis is strictly regulated by hepcidin, a small peptide hormone of 25 amino acids. Hepcidin is predominantly expressed in hepatocytes, and its expression is regulated by the BMP6/ACVR1/SMAD pathway (Figure 1) [28], but it is also produced in other peripheral organs, likely regulating local iron levels [30]. Hepcidin freely circulates in the plasma bound to α2-macroglobulin and is excreted by the kidneys where it is degraded in the glomerulus and/or proximal tubules [31]. Hepcidin levels are inversely associated with the concentration of plasma iron, which is carried by transferrin [32]. This finding was supported by the studies of Riviera and colleagues who showed that injecting synthetic hepcidin in mice rapidly decreased the serum iron concentration within 1 h, in a dose-dependent manner. Furthermore, serum iron returned to the levels prior to hepcidin injection in 96 h; the delay was attributed to the time required to resynthesize ferroportin [33].

Ferroportin is the only known iron efflux transporter and has a central role in mediating iron export to the plasma and extracellular fluids. Ferroportin is expressed in duodenal enterocytes (regulating dietary iron), macrophages in the spleen where erythrocytes are recycled, and hepatocytes where iron is stored [34]. Most importantly, ferroportin is also expressed in erythroid precursors in the bone marrow [35]. Chronic hepcidin expression leads to iron-restricted anemia (usually microcytic, hypochromic anemia). On the other hand, hepcidin deficiency triggers the accumulation of iron in the liver and other parenchyma, sparring the macrophage-rich spleen [36]. Hepcidin levels are also regulated by inflammatory cytokines, such as IL-6; IL-6 regulates the expression of hepcidin via the JAK-STAT3 pathway, which is hyperactivated in myeloproliferative neoplasms (MPNs) [28,37] and contributes to the increased hepcidin levels detected in primary MF patients [38].

## 5. Myelofibrosis and Anemia

Myelofibrosis is the most aggressive MPN, a group of bone marrow cancers that also includes essential thrombocythemia and polycythemia vera. MF is primarily characterized by the replacement of normal bone marrow with fibrotic scar tissue, resulting in impaired production of blood cells [39]. These abnormal alterations in the bone marrow induce the common clinical features in MF, namely, anemia, hepatosplenomegaly (primarily splenomegaly), and systemic symptoms (fatigue, night sweats, bone pain, pruritus, and weight loss, among others). In MF, the median life expectancy ranges between 2 and 6 years from the initial diagnosis [40,41]. Several prognostic models that take into consideration demographic, clinical, genetic, and molecular variables are available for prognostication in primary and secondary MF [42]. MF is considered a chronic disease, primarily occurring in the elderly. Management of MF, which is primarily achieved via treatment with JAK inhibitors, aims to reduce spleen size, control symptoms, improve quality of life, and prolong survival. Currently, four Janus Kinase (JAK) inhibitors (ruxolitinib, fedratinib, pacritinib, and momelotinib) that will be discussed in the following sections have received approval from the Federal Drug Administration (FDA) as treatments for MF in the USA. Several new and promising targeted therapies are in advanced clinical development to treat MF [43,44], such as inhibitors of bromodomain and extra-terminal proteins (e.g., pelabresib), B-cell lymphoma 2 (BCL-2)/BCL extra-large proteins (e.g., navitoclax), and murine double minute 2 (e.g., navtemadlin) [45,46,47,48,49]; these inhibitors may expand the landscape of MF treatments in the near future. Younger and fit patients with MF may be advised to undergo hematopoietic stem cell transplant (the only potentially curative approach) if they harbor high molecular risk mutations (*TP53*, *ASXL1*, *EZH2*, *IDH1/2*, *SRSF2*, and *U2AF1*Q157) or have a high risk of leukemic transformation, due to other well-known risk factors (e.g., blast counts > 1–10%, advanced age, severe anemia, and adverse cytogenetics) [50]. 

Generally, anemia is one of the challenging manifestations in the management of MF, related to multifactorial and not fully understood mechanisms, such as increase in bone marrow fibrosis, higher hepcidin levels, ineffective erythropoiesis, and splenomegaly [51]. Ineffective erythropoiesis contributes to MF-related anemia. Treatment with ruxolitinib can worsen anemia resulting in treatment interruption or dose reductions, which limit its efficacy because spleen responses are dose-dependent. Red blood cell (RBC) transfusion dependence and the administration of a lower ruxolitinib dose (<20 mg BID) at baseline and at 3/6 months are considered negative prognostic risk factors regarding overall survival [52,53]. Also, lower Hb levels (10 g/dL) are inversely associated with quality of life and overall survival, and anemia is considered an adverse factor in the prognostic models commonly used in primary MF; the necessity for RBC transfusions further impacts the prognosis negatively [42].

Several strategies are applied to manage low Hb levels in MF patients, such as treatment with lower doses of ruxolitinib or fedratinib; erythropoiesis-stimulating agents, such as danazol; corticosteroids; and immunomodulatory agents; RBC transfusions; and more recently, treatment with momelotinib (approved in September 2023). RBC transfusions remain the standard approach to maintain Hb levels >7–8 g/dL. However, a heavy RBC transfusion burden lowers the patients’ quality of life and increases medical costs [42]; furthermore, management of anemia with frequent RBC transfusions is associated with several side effects, including iron overload. Usually, one unit of transfused blood contains approximately 200–250 mg of iron. Transfusion-dependent (TD) patients who consecutively receive more than 10–20 units of blood in a short time frame have a significant risk to develop complications secondary to iron deposits, including liver diseases (e.g., cirrhosis and hepatic failure), cardiac diseases (e.g., cardiomyopathies, arrhythmia, and congestive heart failure), endocrinopathies (e.g., diabetes mellitus, hypogonadism, hypothyroidism, and hypopituitarism), arthropathy, and alterations in skin pigmentation [54]. Transfusion-dependent MF patients with these diseases [55] may have a higher risk of infections, as it was reported in a real-life study wherein 45% of 106 patients experienced at least one infection, with a markedly higher incidence in TD patients (HR = 2.13, *p* = 0.019) at a median follow-up of 36 months [56].

Deferasirox, a molecule that chelates trivalent (ferric) iron with high affinity and forms a stable complex that is eliminated by the kidneys, was evaluated in 45 MF TD patients who had iron overload in the multicenter retrospective Italian study, IRON-M. The iron chelation response (ICR) was defined as a stable ferritin level below 1000 μg/L or steady reduction ≥ 50% from baseline. Treatment with deferasirox, at a median starting dose of 10 mg/kg/day (IQR 6.25–25.4), was feasible for all enrolled patients. ICR was achieved in 12 (26.7%) patients. Responders had median ferritin levels of 1390 μg/L vs. 2000 μg/L at baseline (*p* = 0.0018), received a lower number of transfused RBC units prior to ICR (16 vs. 31, *p* = 0.032), and required transfusions for a shorter period of time prior to ICR compared to non-responders (8.7 vs. 16.7 months) [57].

## 6. Concurrent ACVR1 and JAK Inhibition in Myelofibrosis

The turning point in the treatment of MF occurred with the advent of JAK inhibitors, which was inaugurated with the FDA approval of ruxolitinib (JAK1/2 inhibitor) in 2011. Ruxolitinib inhibits the hyperactivated JAK/STAT pathway and revolutionized management of MF patients by significantly reducing the spleen size, controlling constitutional symptoms, and prolonging survival; these benefits were demonstrated in the long-term analyses of the pivotal trials COMFORT-I and COMFORT-II [58,59,60], and global real-life studies [61,62,63,64]. Despite being tolerated very well, ruxolitinib is immunosuppressive, can precipitate opportunistic infections, alter cytokine responses [65,66,67], and may impair response to normal immune stimulation (e.g., SARS-CoV-2 vaccinations) [68,69]. Anemia is the most frequent reason leading to dose reduction/interruption of ruxolitinib in MF patients [70]. The REALISE trial, assessing lower starting doses of ruxolitinib (10 mg BID for the first 12 weeks uptitrated to 25 mg BID if possible) in anemic patients with MF, evidenced that the median Hb levels and RBC transfusion requirements remained stable throughout the study [71]. Furthermore, the RUXOREL-MF study demonstrated the importance of dose optimization in maximizing the clinical benefit of ruxolitinib; a dose < 20 mg BID at baseline and at 3 and 6 months was correlated with inferior overall survival [52]. Importantly, analysis of the data from the large expanded-access JUMP study (2233 MF patients, including patients with platelet counts < 100 × 10^9^/L) demonstrated that new or worsening anemia that developed after initiation of ruxolitinib treatment did not affect the efficacy of the agent regarding spleen and symptom responses [72].

In the past few years, a few other JAK inhibitors were approved for treatment of MF in addition to ruxolitinib [73]. Fedratinib, a JAK2 and FMS-type tyrosine kinase 3 (FLT3) inhibitor, was approved by the FDA (2019) and the European Medicines Agency (2021). Fedratinib presents a possible option for MF patients who are refractory/intolerant to ruxolitinib [74]. Importantly, there was a critical unmet need for MF patients with cytopenias prior to the regulatory approval of pacritinib and momelotinib, the preferable non-myelosuppressive JAK inhibitors to treat cytopenic patients with MF [75], because both ruxolitinib [60,76] and fedratinib [77] may worsen cytopenias.

### 6.1. Pacritinib

Pacritinib (JAK2/IRAK1/ACVR1 inhibitor) was approved as a treatment for MF patients with platelets < 50 × 10^9^/L in February 2022. In a retrospective study of the phase 3 PERSIST-2 trial (NCT02055781), with pacritinib, compared to best available therapy in patients with MF, the blood counts remained stable and anemia benefits were noted, including RBC transfusion independence (TI) [78]. A retrospective analysis of the PERSIST-2 trial demonstrated higher spleen responses and total symptom scores with pacritinib compared to lower doses of ruxolitinib (spleen responses to ruxolitinib are dose-dependent) in MF patients with platelet counts below 100×10^9^/L (22% vs. 3%, *p* = 0.02, and 35% vs. 19%, respectively; *p* = 0.11); pacritinib had a similar safety profile with lower doses of ruxolitinib [79]. The underlying mechanism of pacritinib’s anemia benefits had not been elucidated until its inhibitory activity on the ACVR1/SMAD pathway and suppression of hepcidin expression were demonstrated in 2023 [29]. The mean concentration required for 50% inhibition (IC_50_) of ACVR1 was 16.7 nM for pacritinib and 52.5 nM for momelotinib (Table 1); conversely, ruxolitinib and fedratinib did not show inhibitory activity for ACVR1 [29]. Subgroup analysis of the data from the PERSIST-2 trial was performed to evaluate achievement of TI by comparing the pacritinib-treated arm (200 mg BID) with the best available therapy-treated arm through week 24. Based on the Gale criteria (absence of RBC transfusions over a 12-week period) and the criteria of the SIMPLIFY trials that were designed to evaluate momelotinib regarding TI (absence of RBC transfusions and no Hb < 8 g/dL in the prior 12 weeks), a significantly greater number of pacritinib-treated TD patients achieved TI compared to best available therapy (24% vs. 5%, *p* = 0.013 based on the SIMPLIFY criteria; 37% vs. 7%, *p* = 0.001 based on the Gale criteria) [29].

### 6.2. Momelotinib

On September 15, 2023, momelotinib (JAK1/2 and ACVR1 inhibitor) received FDA approval as a treatment for patients with intermediate- or high-risk MF and anemia based on the findings of the phase 3 trials, MOMENTUM (NCT04173494) and SIMPLIFY-1 (NCT01969838). In preclinical studies, momelotinib inhibited ACVR1/ALK2 (IC_50_ = 8.4 nM; Table 1) [80]. Momelotinib acts by inhibiting the hyperactivated BMP6/ACVR1/SMAD pathway and suppressing hepcidin expression; thus, treatment with momelotinib results in higher circulating iron and Hb levels and improved erythropoiesis [28], which leads to significant anemia benefits in MF patients [81]. The anemia benefits of momelotinib were confirmed in the randomized phase 3 SIMPLIFY-1 and MOMENTUM clinical trials. In the SIMPLIFY-1 trial, momelotinib was compared to ruxolitinib in JAK inhibitor-naïve patients with MF and showed non-inferiority in spleen volume reduction ≥ 35% (SVR35; *p* = 0.011) but not in reducing the total symptom score ≥ 50% (TSS50). Importantly, however, momelotinib reduced TD and the overall RBC transfusion rates. After 24 weeks, 66.5% of the MF patients in the momelotinib arm achieved or maintained TI (based on the aforementioned SIMPLIFY-1 criteria), which was higher than the rate in the ruxolitinib arm (49.3%; *p* < 0.001) [82]. Moreover, stratifying the patients according to baseline Hb levels, for Hb < 8 g/dL, 29% of momelotinib-treated patients achieved TI (vs 18% of ruxolitinib); for Hb  < 10 g/dL, the respective proportions were 47% vs. 27%; and for Hb < 12 g/dL, the respective proportions were 62% vs. 37% [83]. The SIMPLIFY-2 trial (NCT02101268), which compared momelotinib to the best available therapy in previously ruxolitinib-exposed MF patients, demonstrated the superiority of momelotinib in achieving TI compared to the control arm (43% of 104 patients vs. 21% of 52 patients, respectively, *p* < 0.002) at week 24 [84]. Finally, in the MOMENTUM trial, which compared momelotinib with danazol (androgen commonly used to treat anemia in MF) in symptomatic prior JAK inhibitor-exposed MF patients with anemia, the TI rate was 30% in momelotinib-treated patients vs. 20% in danazol-treated patients at week 24; and the TI rate increased by 17% vs. 5% in the groups treated with momelotinib vs. danazol, respectively. Moreover, momelotinib achieved SVR35 in 22% of the patients (vs. 3% for danazol) and TSS50 in 25% of the patients (vs. 9% for danazol) at week 24 [85]. Safety and efficacy were also confirmed at 24 weeks, when all the patients initially randomized to the danazol arm crossed over to the momelotinib arm. Regarding TI, responders at any time during the open-label period by week 48 (using the rolling response definition for TI over any 12-week period by week 24) were 52% and 56% of the evaluable patients who continued momelotinib and patients in the danazol arm who crossed over to the momelotinib arm, respectively [85,86]. Regarding TSS improvements at the same time point, 61% of the patients responded in the momelotinib arm since enrollment, and 59% of the patients in the danazol arm who crossed over to momelotinib were responders [86]. Further analysis of the TSS improvements in the MOMENTUM trial demonstrated ameliorated quality of life as well as rapidity and durability of benefits with momelotinib treatment [87]. Importantly, a recent post-hoc–time-dependent analysis of the data from the SIMPLIFY and MOMENTUM trials demonstrated the statistically significant associations of overall survival with TI arising from treatment with momelotinib [88]. Regarding the associations of RBC transfusion status with overall survival, the hazard ratio (HR) was 5.18 (*p* = 0.0017) in the MOMENTUM trial, and the HR ratio was 3.32 (*p* < 0.0001) and 1.87 (*p* = 0.0287) in the SIMPLIFY-1 and SIMPLIFY-2 trials, respectively [88]. In addition, longitudinal assessment of the RBC transfusion intensity in the SIMPLIFY-1 and MOMENTUM trials demonstrated that superior maintenance of RBC transfusion intensity and zero RBC transfusion status was achieved with momelotinib compared to ruxolitinib in the SIMPLIFY-1 trial; also, momelotinib was superior in decreasing the RBC transfusion burden compared to danazol in the MOMENTUM trial [89].

### 6.3. Jaktinib

Jaktinib (JAK1/2/3 and ACVR1 inhibitor), a deuterated isotope of momelotinib, was first evaluated at a single dose (25–400 mg) and multiple ascending doses (up to 200 mg daily) in healthy subjects and did not induce significant adverse events [90]. Subsequently, in the phase 2 trial in JAK inhibitor-naïve MF patients, 54.8% (34/62) and 31.3% (15/48) of the patients achieved SVR35, and 69.6% (39/56) and 57.5% (23/40) experienced TSS50, in the jaktinib-treated cohorts with 100 mg BID and 200 mg QD jaktinib (*p* < 0.05), respectively. These results were associated with an increase in Hb levels ≥2 g/dL from baseline in 35.6% (21/59) of the patients with Hb ≤ 10 g/dL [91]. In the phase 2b trial enrolling MF patients intolerant to ruxolitinib, 13 (42%) of 31 patients with baseline Hb ≤ 10 g/dL reported ≥20 g/dL Hb increase, and 11 (58%) of 19 TD patients had a 50% reduction in the frequency of RBC transfusions (1 patient became TI) [92]. An interim analysis of the data from the phase 3 trial (NCT04617028) evidenced that 71% of the jaktinib-treated TD patients and 40% of the hydroxyurea-treated TD patients achieved a ≥50% decrease in RBC transfusions by week 24, and one patient achieved TI in the jaktinib arm. Overall, 39.3% of the jaktinib-treated patients and 15.4% of the hydroxyurea-treated patients with baseline Hb ≤ 10 g/dL experienced an increase in Hb ≥ 2 g/dL [93]. Updated results on this ongoing phase 3 trial assessing jaktinib are expected in due time.

**Table 1 cancers-16-00154-t001:** Approved ACVR1 inhibitors and other ACVR1 inhibitors in clinical development for the treatment of anemia in MF patients.

Agent	Study Phase	Indication	Inhibitory Effects
Pacritinib	Approved by the FDA based on the phase 3 PERSIST-2 trial (NCT02055781)	Adults with intermediate- or high-risk MF and platelet counts <50 × 10^9^/L	Mean IC_50_ (ACVR1) = 16.7 nM [28] Inhibitor of JAK2, ACVR1, IRAK1, and FLT3
Momelotinib	FDA-approved based on the phase 3 MOMENTUM (NCT04173494) and SIMPLIFY-1 (NCT01969838) trials	Adults with intermediate- or high-risk MF (primary or secondary) and anemia	Mean IC_50_ (ACVR1) = 52.5 nM [28] IC_50_ (ACVR1) = 8.4 nM [80] Inhibitor of JAK1/2 and ACVR1
Jaktinib	Evaluated in a phase 3 trial in comparison to hydroxyurea (NCT04617028)	In clinical development for MF treatment	IC_50_ (ACVR1) values were not determined. JAK1/2/3 and ACVR1 inhibitor [91]
Zilurgisertib	Evaluated in a phase 1/2 trial as a monotherapy or in combination with ruxolitinib (NCT04455841)	In clinical development for MF treatment	IC_50_ (ACVR1) = 15 nM [94] ACVR1 inhibitor

Abbreviations: ACVR1: activin receptor type I; FDA: Federal Drug Administration; FLT3: fms-like tyrosine kinase 3; IC_50_: half-maximal inhibitory concentration; JAK: Janus kinase; IRAK1: interleukin 1 receptor associated kinase 1; MF: myelofibrosis.

## 7. ACVR1 Inhibitor in Development for Myelofibrosis

Zilurgisertib, a selective oral ACVR1/ALK2 inhibitor, is currently being studied as a monotherapy or in combination with ruxolitinib in a phase 1/2 dose-escalation/expansion trial in patients with MF (NCT04455841). It was recently determined that zilurgisertib had an IC_50_ (ACVR1/ALK2) = 15 nM (Table 1) [94]. Enrolled patients (TD or symptomatic for anemia) were treated with zilurgisertib alone or in combination with ruxolitinib [95,96]. Hepcidin levels (secondary endpoint) decreased in both cohorts after zilurgisertib treatment in a dose-dependent manner, and the maximum decrease of hepcidin levels was noted 6–8 h post zilurgisertib dosing [95]. Hemoglobin increased by >1.5 g/dL from baseline in both groups, and zilurgisertib was well tolerated. The previous findings suggest the promising therapeutic activity of zilurgisertib regarding anemia in MF patients.

## 8. TGF-β Ligand Traps as Treatments for Anemia

Activin receptor type II ligand traps have also been developed to sequester members of the TGF-β superfamily ligands, which inhibit late-stage erythropoiesis. Sotatercept is a novel fusion protein, first-in-class ACVR2A ligand trap that was studied in a phase 2 clinical trial in MF patients with anemia (NCT01712308) [97]. Luspatercept is a recombinant fusion protein (ACVR2B ligand trap) that sequesters TGF-β superfamily members and prevents them from binding to ACVR2B, leading to inhibition of the SMAD2/3 pathway and promoting late-stage erythropoiesis [98]. Luspatercept, a recombinant fusion protein that decreases SMAD2/3 signaling, promotes erythroid maturation and improves anemia. Luspatercept was approved as a treatment for anemia in patients with very low- to intermediate-risk myelodysplastic syndromes (MDS) with ring sideroblasts or MDS/MPN with ring sideroblasts and thrombocytosis (April 2020); lower-risk MDS and requirement for RBC transfusions (August 2023); and β-thalassemia with necessity for RBC transfusions (November 2019) [99]. Future treatments of MF patients with anemia may include regimens combining luspatercept with JAK inhibitors. A published case report showed that the aforementioned combinations could be a viable option for patients with MF [100]. Currently, luspatercept is being evaluated in the ongoing registrational phase 3 clinical trial, INDEPENDENCE (NCT04717414), in anemic MF patients who are concurrently treated with ruxolitinib and required 4–12 RBC transfusions in the 12 weeks prior to enrollment. The results of the phase 2 clinical trial (NCT03194542) that evaluated the safety and efficacy of luspatercept in anemic MF patients (TD or TI) showed significant improvements in anemia and transfusion burden [101].

## 9. Conclusions

ACVR1 inhibition represents a viable target in the management of anemia associated with myelofibrosis. Combining ACVR1 and JAK inhibitors is a promising strategy to target biological pathways that are critical in MF with symptomatic anemia. Dual JAK1/2 and ACVR1 inhibition, such as in the case of momelotinib, provides enhanced therapeutic efficacy in MF given that the cardinal features of the disease (anemia, splenomegaly, and systemic symptoms) are concurrently treated with a single agent. Regulatory approval of momelotinib and pacritinib provided treatment options for the needs of MF patients with cytopenias. Rational combinations of ACVR1 inhibitors or ACVR2B ligand traps, such as zilurgisertib or luspatercept, respectively, with JAK1/2 inhibitors (for example, ruxolitinib) also are promising strategies to treat anemic patients with MF. Additional clinical trials evaluating other promising novel agents and studies reporting real-world evidence are necessary to validate the therapeutic potential of novel ACVR1 inhibitors, ACVR2 ligand traps, or other agents to treat anemia in MF and pave the way to broader clinical applications and paradigm changes in the treatment of MF. For example, DISC-0974, a first-in-class human anti-hemojuvelin (HJV) monoclonal antibody, is currently being evaluated in a phase 1b/2a trial in patients with MF and anemia (NCT05320198). HJV is a co-receptor of the BMP family ligands that regulates hepcidin expression and iron metabolism. HJV demonstrated dose-dependent suppression of hepcidin and an increase in serum iron and Hb levels in healthy volunteers [102]. Novel agents along with the recently approved medications are poised to expand the armamentarium of treatments for anemia associated with MF and improve the outcomes and quality of life in MF patients.

## Figures and Tables

**Figure 1 cancers-16-00154-f001:**
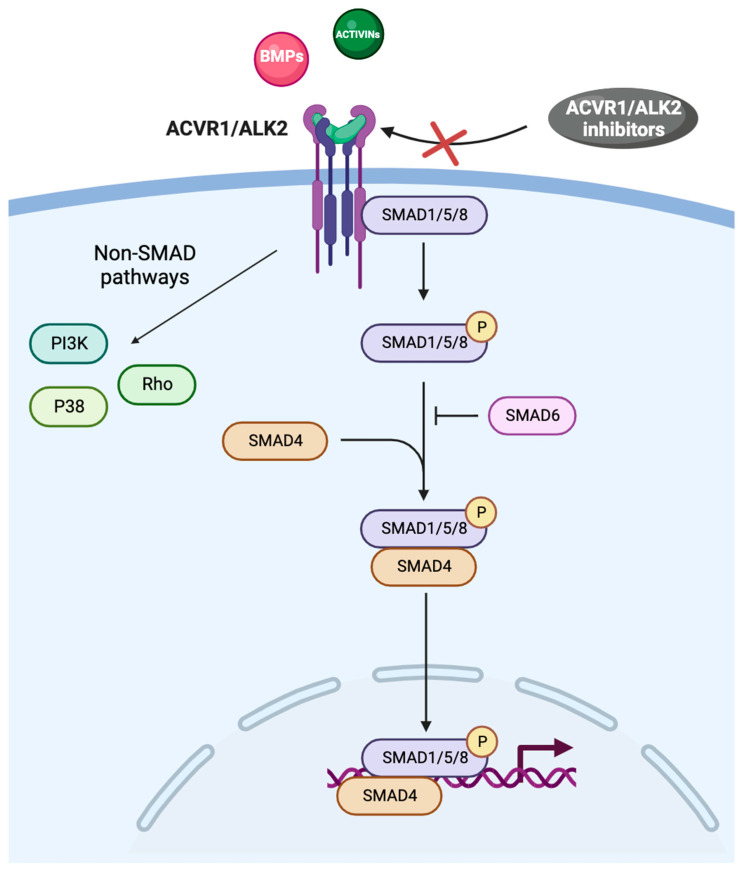
The BMP/ACVR1/SMAD pathway. Abbreviations: ACVR1: activin receptor type I; ALK2: activin receptor-like kinase-2; BMPs: bone morphogenic proteins; PI3K: phosphatidyl-inositol 3-kinase; SMAD: small-mothers-against decapentaplegic.

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
