# Peer review of "ACVR1: A Novel Therapeutic Target to Treat Anemia in Myelofibrosis"

_cancers, 2023, doi:10.3390/cancers16010154_

Round 1
Reviewer 1 Report
Comments and Suggestions for Authors
Thanks so much to the authors for writing and sharing this very nice review article. It is very well-written, easily readable, and certainly relevant right now. I have a few comments and suggestions, all of which are minor:
Line 108—The mention of Indian hedgehogs is confusing to me—how does this differ from Shh? Most clinically-oriented readers will be aware of Shh, but not Indian hedghogs, so a bit more clarification might help.
Line 177 – would mention United States (only 1 JAKi in most other countries)
Line 188 – would mention more about mechanisms of anemia in MF, which are multifactorial (splenomegaly, etc)
Line 202 – RBC transfusions and QOL, need citation
Line 203-- continuous hypertransfusion? Would simplify terminology
Lines 257-264 – PERSIST-2 described twice
Line 273 mentions the SIMPLIFY trial without context—would suggest explaining this study at first mention.
Line 303/304 would rephrase to “prior JAK inhibitor exposed”
DISC-0974 drug mentioned in intro and conclusion, but seems like it deserves more discussion in section 6.4. No clinical data to discuss, but lots of preclinical stuff and a bit unique mechanistically.
Section 7 – Luspatercept now approved for pts with lower-risk, transfusion-dependent MDS in the frontline setting as well. Maybe mention why it’s applicable to MDS since that disease hasn’t been discussed at all.
How about mentioning the situation of hepcidin mimetics (Rusfertide, silence drug) in polycythemia vera? Similarly, is it worth mentioning IRIDA as mechanistically this disease is a good example of the clinical relevance of this mechanism? Just suggestions—feel free to ignore if you prefer!
Author Response
REVIEWER 1
|
Reviewer observation |
Author response |
|
Thanks so much to the authors for writing and sharing this very nice review article. It is very well-written, easily readable, and certainly relevant right now. I have a few comments and suggestions, all of which are minor:
Line 108—The mention of Indian hedgehogs is confusing to me—how does this differ from Shh? Most clinically-oriented readers will be aware of Shh, but not Indian hedghogs, so a bit more clarification might help. Line 177 – would mention United States (only 1 JAKi in most other countries) Line 188 – would mention more about mechanisms of anemia in MF, which are multifactorial (splenomegaly, etc) Line 202 – RBC transfusions and QOL, need citation Line 203-- continuous hypertransfusion? Would simplify terminology Lines 257-264 – PERSIST-2 described twice Line 273 mentions the SIMPLIFY trial without context—would suggest explaining this study at first mention. Line 303/304 would rephrase to “prior JAK inhibitor exposed” DISC-0974 drug mentioned in intro and conclusion, but seems like it deserves more discussion in section 6.4. No clinical data to discuss, but lots of preclinical stuff and a bit unique mechanistically. Section 7 – Luspatercept now approved for pts with lower-risk, transfusion-dependent MDS in the frontline setting as well. Maybe mention why it’s applicable to MDS since that disease hasn’t been discussed at all.
|
Thanks for your comments and suggestions. They were very interesting and we worked on the paper according to them, in order of improving the readability and interest of the manuscript. Specifically: - We simplified the Indian hedgehogs/Shh concept - We expanded the concept of DISC-0974, but we prefer to leave the discussion in the future direction section, as it isn’t a direct inhibitor of ACVR1, as the paragraph title refer - We agree with the remaining suggestions, modifying the text according to |
|
How about mentioning the situation of hepcidin mimetics (Rusfertide, silence drug) in polycythemia vera? Similarly, is it worth mentioning IRIDA as mechanistically this disease is a good example of the clinical relevance of this mechanism? Just suggestions—feel free to ignore if you prefer! |
Thank you for your observation, we added the discussion regarding it in the manuscript, mentioning these concepts |

Reviewer 2 Report
Comments and Suggestions for Authors
In this review by Duminuco et al, the authors detail ACVR1 as a promising target for anemia in the setting of myelofibrosis. The draft is expertly prepared although I have some content requests which I think would enhance the clinical relevance for practicing hematologist.
- The discussion of FOP is interesting, but should not be so detailed. A few sentences describing this disease suffices but it is not relevant for the hematology community. Would recommend removing or sharply limiting this.
- What is most missing, in my opinion, is the authors opinion on how to choose between current and future ACVR1 inhibitors. How would you choose between momelotinib and pacritinib? What is the hypothetical benefit of zilurgisertib over JAK inhibitors with ACVR1 inhibition?
- The authors should expand on combination between ACVR2 ligand traps and ACVR1 inhibition as this is provocative and of considerable interest for the future therapeutic direction of anemic myelofibrosis
- Minor: the units for hemoglobin should be consistent throughout (g/dL vs g/L)
Author Response
REVIEWER 2
|
Reviewer observation |
Author response |
|
In this review by Duminuco et al, the authors detail ACVR1 as a promising target for anemia in the setting of myelofibrosis. The draft is expertly prepared although I have some content requests which I think would enhance the clinical relevance for practicing hematologist |
Thank you for the comment, attached below our response to your suggestions |
|
- The discussion of FOP is interesting, but should not be so detailed. A few sentences describing this disease suffices but it is not relevant for the hematology community. Would recommend removing or sharply limiting this |
Thanks for your comment, we reduced the discussion concerning FOP |
|
- What is most missing, in my opinion, is the authors opinion on how to choose between current and future ACVR1 inhibitors. How would you choose between momelotinib and pacritinib? What is the hypothetical benefit of zilurgisertib over JAK inhibitors with ACVR1 inhibition?
|
We reported this concept in the conclusion section, evidencing the future landscape for this molecules |
|
- The authors should expand on combination between ACVR2 ligand traps and ACVR1 inhibition as this is provocative and of considerable interest for the future therapeutic direction of anemic myelofibrosis
|
Thanks for your suggestion, we improved the manuscript stressing the concept of ACVR1/2 ligand traps association |
|
- Minor: the units for hemoglobin should be consistent throughout (g/dL vs g/L) |
We corrected the hemoglobin unit, standardizing with the use of g/dL |
